# WINNER'S CURSE?
## ON PACE, PROGRESS, AND EMPIRICAL RIGOR

**D. Sculley, Jasper Snoek, Ali Rahimi, Alex Wiltschko**
{dsculley, jsnoek, arahimi, alexbw}@google.com
Google AI

### ABSTRACT

The field of ML is distinguished both by rapid innovation and rapid dissemination of results. While the pace of progress has been extraordinary by any measure, in this paper we explore potential issues that we believe to be arising as a result. In particular, we observe that the rate of empirical advancement may not have been matched by consistent increase in the level of empirical rigor across the field as a whole. This short position paper highlights examples where progress has actually been slowed as a result, offers thoughts on incentive structures currently at play, and gives suggestions as seeds for discussions on productive change.

## 1 INTRODUCTION: COMPETITION MINDSET?

The field of ML has seen extraordinary progress, with error rates dropping by an order of magnitude in image classification in the last decade, and similar levels improvement in application areas ranging from machine translation to computational biology to autonomous vehicles to playing chess, Go (Silver et al., 2017), and video games (Mnih et al., 2015). The pace of this progress has grown in a research and publication culture that emphasizes *wins*, most often demonstrating that a new method beats previous methods on a given task or benchmark. It is a truism within the community that at least one clear win is needed for acceptance at a top venue. Yet, a moment of reflection recalls that the goal of science is not wins, but knowledge.

In the context of contemporary ML – which has come to emphasize deep learning, non-convex optimization, and other methods that yield impressive empirical results but are difficult to analyze theoretically – we believe empirical rigor is more important now than ever before. Yet we have not, on the whole, seen a material increase in the standards for empirical analysis and empirical rigor across the field. This paper is hardly the first to voice such questions (e.g. Rahimi & Recht). and while this paper refers to the field as a whole, there are many individual papers included in the references that give important advances in empirical analysis or understanding. Still, we do hope to add usefully to the conversation.

In particular, we begin with an informal meta-analysis based on several recent papers each of which individually has found evidence in which a possible lack of rigorous standards for empirical work in the field has led to delays, reduced progress, or churn in various topical sub-fields including sequence-to-sequence learning (Melis & Chris Dyer), reinforcement learning (Henderson et al., 2017), GAN's (Lucic et al., 2017), and bayesian deep learning. Understanding that researchers in the field are well intentioned, we look for *structural* incentives and mechanisms that may be responsible for some of these effects. Finally, we offer suggestions towards structural changes that may better incentivize and reward a higher level of empirical rigor in the field at large.

## 2 HIGHLIGHTING CASE STUDIES FROM THE LAST YEAR

Looking over papers from the last year, there seems to be a clear trend of multiple groups finding that prior work in fast moving fields may have missed improvements or key insights due to things as simple as hyperparameter tuning studies or ablation studies.

Lucic et al. (2017) conducted a large scale empirical comparison of recent innovations in generative adversarial networks. A main finding was that most recent methods would reach similar scores with sufficient hyperparameter optimization.

Henderson et al. (2017) demonstrated that they could beat a host of recent methods in sequence-to-sequence learning to get state-of-the-art performance on the hotly contested Penn Treebank dataset simply by doing better hyperparameter tuning on the baseline LSTM.

Vaswani et al. (2017) effectively performed an ablation study on exotic encoder-decoder style networks with attention and demonstrated that one could perform as well or better using just the attention module.

Rikelme et al. (2018) compared a variety of recent approaches for decision making using approximate inference in Bayesian deep neural networks. They found that in decision making tasks, many recently proposed methods struggled to outperform simple baselines.

Henderson et al. (2017) reviewed reproducibility in deep reinforcement learning and found significant variability between baseline implementations across recent work.

Together, these papers show several areas where seemingly fast progress was perhaps slower than it could have been if the field had enforced higher levels of empirical rigor.

## 3    INCENTIVES AND CONDITIONS

Consider the dramatic growth in machine learning in the following areas:

- Publicly available data sets. Kaggle now hosts more than 10,000 public data sets.
- Availability of cheap compute, along with a corresponding increase in the availability of large scale computing resources such as cloud.
- The number of researchers working in the field, creating opportunities for large scale collaborations.
- The rise of open source ML platforms such as TensorFlow and PyTorch, and the resulting spread of open source code and models.

History will tell whether these areas of growth will yield an increase in useful results, but on their face, these factors ought to foster ever stronger empirical work. But each of these advances comes with a countervailing force that can hamper the pace of progress.

- Empirical studies have become challenges to be "won", rather a process for developing insight and understanding. Ideally, the benefit of working with real data is to tune and examine the behavior of an algorithm under various sampling distributions, to learn about the algorithms strengths and weaknesses, as one would do in controlled studies.
- The price of compute is relative. Large research groups (often based in industry) may have the resources, for example, to tune models on 450 GPUs for 7 days (Esteban Real, 2018), but individual researchers may be harder pressed. This may require larger collaboration groups, as is often seen in fields such as physics.
- As the number of participants in the field grows, the acceptance rate at top venues seem to have remained constant (Mozer, 2017). Furthermore, because it takes years to train good reviewers, the number of skilled reviewers will necessarily lag behind. Because publication has a strong impact on career growth, there is an increased fear of getting scooped by a would-be competitor hoping to quickly plant a flag in a territory. This fear may disincentivize taking time to perform fine-grained empirical analysis, especially when page limits restrict the ability to include additional empirical depth.
- When many reseachers work in parallel in a problem or a related set of problems, the field can experience the issues of multiple hypothesis testing even when individuals are taking pains to avoid them.

## 4    IDEAS FOR CHANGE

While there are no easy fixes, we take inspiration from papers such as Soergel et al. that have helped create change structures and incentives in the field, and offer a suggestions as the starting points for further conversation.

**Standards for Empirical Evaluation**    In addition to current practice, we feel the following additional standards should be encouraged, rewarded, and ultimately required in empirical work:

- **Tuning Methodology** Tuning of all key hyperparameters should be performed for all models including baselines via grid search or guided optimization (*e.g.* such as Golovin et al. (2017); Snoek et al. (2012); Hutter et al. (2011)), and results shared as part of publication.

- **Sliced Analysis** Performance measures such as accuracy or AUC on a full test set may mask important effects, such as quality improving in one area but degrading in another. Breaking down performance measures by different dimensions or categories of the data is a critical piece of full empirical analysis.

- **Ablation Studies** Full ablation studies of all changes from prior baselines should be included, testing each component change in isolation and a select number in combination.

- **Sanity Checks and Counterfactuals** Understanding of model behavior should be informed by intentional sanity checks, such as analysis on counter-factual or counter-usual data outside of the test distribution. How well does a model perform on images with different backgrounds, or on data from users with different demographic distributions?

- **At Least One Negative Result** Because the No Free Lunch Theorem still applies (Wolpert & Macready, 1997), we believe that it is important for researchers to find and report areas where a new method does *not* perform better than previous baselines. Papers that only show wins are potentially suspect and may be rejected for that reason alone.

**Sharing Experimental Notes and Records**    Our experience is that ML researchers often avoid the practice from other fields of recording all results as they happen in physical *notebooks*. We suggest requesting in CFP's detailed (and time-stamped) notes in an electronic doc on all experiments run during the research for a paper. These help trace the course of development, exploration, and conclusions; and they can counteract issues of multiple hypothesis testing and post-hoc explanations.

**Options for Paper Structure**    In a field known for innovation, it is perhaps surprising that our primary medium of archival communication are papers that are still optimized for being printed *on paper*. Alternative paper formats, including smart notebooks like iPython and Colaboratory [1] that include code, data, and analysis along with text should be first-class publication media.

Conference paper page limits at conferences restrict the ability to report more complete empirical analysis, which internally we have found to often take many pages.. Once these limits were due to physical printing costs, but now reviewer bandwidth is the primary constraint. As one idea, we offer the suggestion of flexible page limits for empirical results, with the caveat that authors perform one additional conference review for each additional page of empirical results as a "co-pay" to avoid abuse, coupled with appropriate standards on review quality.

**Collaboration and Credit Assignment**    To achieve dramatically more complete empirical evaluation and analysis is fundamentally more work, and will likely be best achieved by larger groups of collaborators. Incentivizing such collaborations is difficult in a field in which credit assignment is inferred via the low-bandwidth signal of author ordering. We suggest enabling an appendix in each paper briefly summarizing each author's contribution as one possible solution.

**Standards for Reviews and Reviewers**    Review quality is a key factor to raising the bar on empirical rigor in the field. We suggest helping reviewers and Area Chairs enforce higher review standards by creating better tooling for reviewer such as the ability to add line-item comments directly in text, creating more complete rubrics for review, and reserving a large number registrations to hard-to-attend conferences for Area Chairs to award to strong reviewers.

**Options for Venues**    Currently, conference paper acceptance rates are linked to the physical size of available venues. We suggest working more creatively with alternative media (including video and video conferencing) to flexibly create additional opportunity to accept papers that focus on issues other than wins, such as in depth meta-analyses commonly found in other fields.

---

[1] https://colab.research.google.com

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
