# OpenReview forum: "Winner's Curse?  On Pace, Progress, and Empirical Rigor"
_ICLR.cc/2018/Workshop — Accept_

### Official Review · AnonReviewer1 · 2018-03-01
**Fantastic paper for discussion and basically I could not agree more**

**Rating:** 10
**Confidence:** 5

**Review:**

The issues presented here are very important for the community to discuss.   This paper extends on, concertizes and substantiates with references  of key ideas of the talk Ali Rahimi gave at NIPS in his acceptance of the Test of Time award with Ben Recht.   That talk was highly influential in that talk with the addition of references , more substance and a path forward.  Most notable, Yann LeCunn has come out to contradict some of these ideas, inspiring a debate on Facebook.   Great.  Let's talk more.  The community needs to have these discussions and this is an excellent position paper that is sure to have impact.

Here are the points I like, that I am glad are being documented in this paper and that I think should be recorded and discussed in this forum:

>The price of compute is relative. Large research groups (often based in industry) may have
>the resources, for example, to tune models on 450 GPUs for 7 days (Esteban Real, 2018),
>but individual researchers may be harder pressed.

I am glad someone wrote this down.  At ICML 2016 in New York I remember one presenter (on deep reinforcement learning) saying to the audience (and this is a likely incorrect paraphrased memory) "this step you can do on your home PC, and this one, this step basically brought down one of Google's data centers, I don't recommend trying this one at home <laughter> " and I remember thinking to myself how great it would be to see that kind of honest commentary in the actual paper.

Much in the spirit of this paper and the OpenReview protocol, the more rigorous we can be and the more truly honest we can be about our methods (requirements, limitations) the better it will be for the field overall.

With respect to the recommendations, I am a particular fan of: At Least One Negative Result and Tuning Methodology.  Although they are all probably good ideas, these in particular resonate.

We should all be talking about: "Alternative paper formats, including smart notebooks like iPython and Colaboratory 1
that include code, data, and analysis along with text should be first-class publication media."  In my opinion these should be first class and paper alone can go back to coach.

I would also like to applaud the authors for their appeal for "Standards for Reviews and Reviewers - Review quality is a key factor to raising the bar on empirical rigor in the field."  I am hoping that tools like OpenReview will help here as more popular works will hopefully attract more discussion and criticism.  As part of the "Winner's Curse" - review quality has (almost necessarily) gone down due to volume.  With record numbers of submissions at NIPS and ICML there has been a crisis in finding a sufficient number of qualified reviewers for papers.  The NIPS consistency experiment was an incredibly bold move to evaluate the review process and take steps to improve it, but implemented changes, for the better are up against a flood of submissions.  Also with the amount of review requests many of us receive the amount of time we can give each paper drops.  "The ability to add line-item comments directly in text" would be excellent as we could all better contribute what we can to the process.  Imagine being able to see a research paper like a word document with all the mark-up and edits from your collaborators (maybe one at a time).  Would we even get to the point of "publishing" rejected papers where critical flaws in proof s are revealed and the reviewers who found them are rewarded.  In this case the authors would remain anonymous and the reviewer would be named.

Furthermore: "We suggest working more creatively with alternative media (including video and video conferencing) to flexibly create additional opportunity to accept papers that focus on issues other than wins, such as in depth meta-analyses commonly found in other fields." - also I could not agree more.  It would be great to not have to travel as much, to deal with visa issues, to be able to have extended conversations.  Instead of repeating the opening comments to your poster, these could be recorded - imagine every poster presenter giving their generic intro presentation on a video and submitting that to the conference website.  Then you could ask questions in the comments or even arrange a video chat - and maybe if you are daring - record the chat and add it to the conversation.

Overall, this paper is very good and should be accepted.  Hopefully there will be other reviews that strongly disagree and add more perspectives to the conversation, but I would submit that such disagreement would only strengthen the reasons I think this paper should be published: it provides excellent talking for the community.

---

### Official Review · AnonReviewer3 · 2018-03-10
**Important questions are raised but few answers are given**

**Rating:** 7
**Confidence:** 3

**Review:**

As we have seen in the replication crisis in many scientific areas, good incentives and good institutional structures are critical to scientific work.  This paper presents a welcome perspective and will yield a welcome discussion on this topic.

I would say that the paper raises important questions and points of discussion but offers few tenable answers.

This is an interesting proposition: "Papers that only show wins are potentially suspect and may be rejected for that reason alone."

But laying out standards for best practice is different from creating institutional incentives that support these best practices.

We need conferences and journals to enforce what we think are the best practices in the field.  E.g., the Journal of Experimental Psychology has adopted stringent requirements for papers submitted to the journal.  Your paper is not accepted for review unless it meets the basic standards outlined.

The Options for Paper Structure seem a bit half-baked. Submitting python notebooks along with a paper seems fine, but there is definitely value to the written word. Many people will not be interested enough in your paper to dig into the details, but some will, and we should accommodate both types of readers as best as possible.

No reasonable solutions are offered for the credit assignment problem. Granted, it's a hard problem.

I think it's good to discuss these problems, but more work is needed to find viable interventions.

---

### Official Review · AnonReviewer2 · 2018-03-14
**Much needed position paper**

**Rating:** 9
**Confidence:** 5

**Review:**

This is a nicely written position paper that points out many important aspects of the current  state of empirical rigor and analysis present in the ML field. Authors highlight very important (if not critical for the true advancement of science) aspects (mainly issues) related to the empirical studies conducted and reported in the recent past.
Authors emphasize that the lack of standards in empirical rigor is a most recent trend. It would be good to talk about how recent is most recent.
In the last part of the paper authors give suggestions on how improvement could be made. I don't certainly agree with some of the ideas in full. In general I find it appealing that the authors list and describe specific ideas through which improvements could be made. As they point out these are suggestions for further discussions.
For example, the idea of enabling appendix where each author's contribution would be summarized. Authors could contribute on many levels including discussions where ideas could be spurred which won't necessarily be implemented by the same person that originally came up with that idea. I doubt that could be easily summarized.

I find it especially important that the authors emphasize the standards for reviews and reviewers. In regards to enforcing higher review standards I would encourage the authors to discuss the idea of having a proper "training" of reviewers that goes beyond having a single "guidelines" page provided by the conference organizers. Perhaps even a short conference specific online course that each reviewer ought to take prior to reviewing. Such a course would ensure that all reviewers would properly familiarize themselves and abide to the conference standards. Even if better tools are created for the reviewers there are no guarantees that reviewers would make proper use of the them unless they are properly instructed - an online course could certainly help with that as well.

---

### Public Comment · (anonymous) · 2018-03-01
**thanks for the perspective**

Thanks so much for writing this paper and sharing these perspectives! One important work that aimed to established rigor, but is not mentioned in this paper, is [1]. In this work, the authors show that many of the popular benchmark tasks used in hundreds of deep reinforcement learning papers, can in fact be solved with linear policies or policies using random features. This is probably the perfect illustration of jumping the gun -- trying neural networks without trying linear regression, and also very much in line with what the authors of this ICLR submission are attempting to illustrate.

[1] Rajeswaran et al. Towards Generalization and Simplicity in Continuous Control, NIPS 2017.

---

> ### Public Comment · ~Jasper_Snoek1 · 2018-03-02
> **Thanks for the reference**
>
> Thanks for the feedback.  This seems relevant and we'll add it as a citation in the paper!

---

### Public Comment · ~Frank_Hutter1 · 2018-03-06
**Thanks, these are important topics! Should also mention reproducibility / making available code**

Thanks a lot for writing this paper, I believe it is important that these points are made.

I strongly believe that one additional point should be made, though: reproducibility / making available code.
So much of our algorithms' performance depends on their implementation, and the full details are *never* in the paper, even if the authors have the best of intentions. The only way to ensure reproducible research is therefore to make available code. In contrast to basically all other sciences, in computer science we have a great opportunity: for us, it is trivial to make results reproducible, by simply making code available. (Even in the extreme case of requiring 450 GPUs for a week, there likely exists at least one other group who could directly reproduce the results.)

At the same time, note that the incentive system with respect to releasing code is completely broken.
Take two hypothetical research labs A and B that both work on the same topic. A makes all their code available for reproducibility; B doesn't. This immediately provides B a competitive advantage because they can put all of A's advances into their code, whereas A cannot use any of B's advances, because that would entail reproducing B's work first. Writing a paper combining A+B's work will therefore be trivial for B and almost impossible for A. So, there is no incentive for B to change their policy and make code available, but there is actually incentive for A to change their policy and *stop* making their code available. This is a clear example of the prisoners' dilemma.

Deep learning thrives because a lot of people are still making available code for their research. But unfortunately, most of the big research labs are of type B and don't make their research code available. (They have made a lot of fantastic framework code available, and for that I am very thankful! The argument I'm making here is purely about research code.) The typical argument made by authors at industrial research labs, even those with the best of intentions, is that their research code depends on too many company-internal packages or specific compute infrastructure, and nobody else could get it to run. However, if the incentive system was such that making code available is rewarded, this would change immediately. Say, all of NIPS, ICML, and ICLR decided that they only publish papers for which code is available to reproduce all experiments in the paper. I strongly believe that within a year all research labs would then find ways to make their code available. They simply would have to in order to keep the best researchers. That's the power of incentive systems.

I'm not proposing to do this against companies, but *with them*. I fully understand that industrial research labs need to keep in mind who pays their bills and can't just give all their code away to competitors. But I believe that could be solved by appropriate licenses; e.g., a license that allows free use for non-commercial purposes would fully suffice for reproducibility but still guard against other companies using the code for their products.

I think it would improve the paper to address this issue of reproducibility.
Thanks for an important call for more empirical rigor in the community!

---

> ### Public Comment · ~D._Sculley1 · 2018-03-06
> **thanks for the thoughtful comment**
>
> Thanks for this thoughtful comment.  I think we totally agree here -- releasing research code is a key step, and we should find additional structural incentives to reward (or potentially require) this as part of publication.
>
> For what it's worth, the only additional comment I'd make here is that this may not be enough, as ML code without data is only part of the story.  So in the paper, we prefer to go a step further and ask / reward / require not just code, but also data and analysis that accompany experiments.  This is the motivation behind our suggestion to begin considering alternative paper formats such as smart notebooks (like iPython and CoLaboratory) as first-class publication media.  But I agree more can be said here to emphasize this important point.

---

### Public Comment · (anonymous) · 2018-03-23
**Very nice! Maybe also mention openml.org?**

Very nice paper!

Some of the paper's ideas are very much in line with the open machine learning website openml.org -- maybe a reference could be added? (https://www.openml.org/cite) That website has thousands of datasets available with a nice API (kind of like UCI++), and there are also benchmarking suites and notebooks available for directly comparing a new algorithm's results with previously known ones; e.g., https://arxiv.org/abs/1708.03731


Also, it's not that far fetched for ML conferences to require / reward best practices for reproducibility. E.g., ECML-PKDD allows authors to flag their submissions as "reproducible research", which gives them a bonus, but which requires posting code and data on standard repository hosting services. Probably ICLR, ICML, and NIPS should follow suit, and I hope this paper helps make this happen rather sooner than later.

---

### Decision · Program_Chairs · 2018-03-20
**ICLR 2018 Workshop Acceptance Decision**

**Decision:**

Accept

**Comment:**

Congratulations, your paper was accepted to the ICLR workshop.